# One-Step and Morphology-Controlled Synthesis of Ni-Co Binary Hydroxide on Nickel Foam for High-Performance Supercapacitors

**Xiao Fan, Per Ohlckers *** 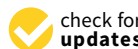 **and Xuyuan Chen ***

Department of Microsystems, Faculty of Technology, Natural Sciences and Maritime Sciences,
University of South-Eastern Norway, Campus Vestfold, Raveien 215, 3184 Borre, Norway; xiao.fan@usn.no
* Correspondence: per.ohlckers@usn.no (P.O.); xuyuan.chen@usn.no (X.C.); Tel.: +47-310-09-315 (P.O.);
+47-310-09-028 (X.C.)

**Abstract:** Ni-Co binary hydroxide grown on nickel foam was synthesized through a facile one-step process for pseudocapacitive electrode application. The morphology of the fabricated binary hydroxide, evolving from nanosheet to nanowire, was highly controllable by tuning the Ni:Co ratio. In systematical electrochemical measurements, the prepared binary material on nickel foam could be employed as a binder-free working electrode directly. The optimal composition obtained at the Ni:Co ratio of 5:5 in integrated nanosheet/nanowire geometry exhibited high specific capacitances of 2807 and 2222 F/g at current densities of 1 and 20 A/g, equivalent to excellent rate capability. The capacitance loss was 19.8% after 2000 cycles, demonstrating good long-term cyclic stability. The outstanding supercapacitors behaviors benefited from unique structure and synergistic contributions, indicating the great potential of the obtained binary hydroxide electrode for high-performance energy storage devices.

**Keywords:** Ni-Co binary hydroxide; one-step; controllable morphology; binder-free; supercapacitors; synergistic contributions

## 1. Introduction

The enormously increasing energy requirements and the limited availability of energy sources motivate the development of efficient and safe energy storage devices [1,2]. Supercapacitors, as next-generation high-performance energy storage devices, significantly make up the shortfall of conventional physical capacitors and batteries, and are extremely suitable for superior power density and high charge/discharge rate required applications [3–5], such as electrical vehicles and stop-go driving models [6,7]. In addition, supercapacitors also possess an ultralong service life of up to 10,000 cycles and excellent operating temperature adaptability from −40 °C to +70 °C [8,9]. Accordingly, supercapacitors have sparked extensive interest in recent years. Two types of supercapacitors, divided by energy storage mechanisms, are termed as electric double-layer capacitors and pseudocapacitors, respectively [10,11]. Currently, the electric double-layer capacitors, by operating reversible ions' adsorption/desorption to store energy, suffer from lower specific capacitance [12,13], which fails to meet the ever-growing demand. In contrast, pseudocapacitors contribute to higher specific capacitance by reversible faradaic redox reactions [14,15] and have become the research hotspot in supercapacitors fields in past decade.

Among various transition metal oxides/hydroxides for the electrode materials of pseudocapacitors, $Ni(OH)_2$ and $Co(OH)_2$, in virtues of definite redox transitions, superior theoretical specific capacitances (ca. 3750 F/g for $Ni(OH)_2$ and ca. 3460 F/g for $Co(OH)_2$), earth-abundant resources, and environmental friendliness [16,17], have attracted much attentions. For instance, the $Ni(OH)_2$ microspheres,

synthesized by Du et al., presented the specific capacitance of 1280.9 F/g at current density of 0.5 A/g [18]. Yin et al. reported the $Co(OH)_2$ nanoflakes delivering 1636 F/g at 0.5 A/g [19]. However, the achieved specific capacitances by single $Ni(OH)_2$ or $Co(OH)_2$ so far are far lower than the theoretical values, consequently hampering their practical use. To address the issue, novel binary Ni-Co hydroxide was studied and has been shown to outperform the corresponding single hydroxide, owing to unparalleled advantages, such as stronger layered orientation, reduced resistance, increased active sites generated by valence interchange or charge hopping between cations, synergistic redox reaction, moderate redox potential, and decreased redox peak potential difference [20–28]. Unfortunately, even though numerous efforts were devoted, the specific capacitances up to now are still unsatisfied, which are mainly caused by undesirable morphology [29–33] and involved binders [34–38]. Moreover, with respect to high rate capability, cycling stability, good mass loading, and facile method, previous progress to date rarely shows all of these characteristics. For example, the Ni-Co double hydroxide nanosheets prepared by Chen et al. reached ultrahigh 2682 F/g at 3 A/g, but the capacitance loss was more than 35% at 20 A/g [39]. Similarly, the Ni-Co double hydroxides microspheres by Tao and co-workers displayed 2275.5 F/g at 1 A/g, nevertheless the capacitance sharply faded to 1007.8 F/g at 25 A/g [40]. Besides, Zhang et al. developed flower-like Ni-Co binary hydroxides, in which only 73.8% capacitance could be maintained after 3000 cycles [41]. Yang and co-workers fabricated Ni-Co hydroxide nanostructures, which demonstrated 1760 and 1468 F/g at 1 and 20 A/g, as well as 87.3% capacitance retention after 2000 cycles, whereas remarkable criteria were obtained at an exaggerated low mass loading of 0.23 mg/cm$^2$ [42]. In addition, the multi-step and complex routes employed in certain cases further hindered their commercial application. Hence, it is still a challenge to get the utmost out of the binary system for the application of high-performance supercapacitors with promising commercial prospect.

In this paper, we report a novel approach to directly grow Ni-Co binary hydroxides at different Ni:Co ratios on nickel foam (NF), a substrate with a series of fulfilling features, such as polyporous structure, high surface area, and low resistivity. The optimal electrode has numerous advantages, such as being binder-free, having integrated geometry, moderate redox potential, overlapping redox peaks, and feasible synergistic effect, therefore displaying outstanding overall supercapacitor performances. The simple one-step method further manifests the bright outlook of this study in both research and commercial fields.

## 2. Materials and Methods

### 2.1. Preparation of Ni-Co Binary Hydroixde on Nickel Foam

Ni-Co binary hydroxide on nickel foam was fabricated by a facile one-step solvothermal reaction. Nickel foam (1 cm × 1 cm) was pretreated with 6 M HCl, deionized water, and ethanol, sequentially. Then, a precursor solution containing the desired molar mixture of $Ni(NO_3)_2 \cdot 6H_2O$, $Co(NO_3)_2 \cdot 6H_2O$ (the feeding concentration ratios of $Ni^{2+}:Co^{2+}$ were 9:1, 7:3, 5:5, 3:7, and 1:9, respectively, and total molar of cations was 5 mmol), 2 mmol $NH_4F$, 6 mmol $CO(NH_2)_2$, and 50 mL deionized water was prepared. After vigorous stirring for 10 min, the solution and nickel foam were transferred to a Teflon-lined stainless-steel autoclave and maintained at 120 °C for 8 h. The final products were rinsed with deionized water after cooling down to room temperature naturally. For convenient clarification, the samples were labeled as $Ni_{1-x}Co_x(OH)_2$ (x = 0.1, 0.3, 0.5, 0.7, and 0.9). The loading mass on the nickel foam was approximately 1.5 mg/cm$^2$.

### 2.2. Material Characterizations

The surface morphological feature was characterized by a scanning electron microscope (SEM, SU8230, Hitachi, Tokyo, Japan) operated at 10 kV. The crystalline structure was investigated via X-ray powder diffraction (XRD, EQUINOX 1000, ThermoFisher, Waltham, America) with Cu Kα radiation (λ = 0.15406 nm). X-ray photoelectron spectroscopy (XPS, ESCALAB 250Xi, ThermoFisher, Waltham, America) under monochromatized Al Kα excitation was adopted to reveal the chemical status.

### 2.3. Electrochemical Measurements

The electrochemical performances were evaluated throughout cyclic voltammetry (CV), galvanostatic charge/discharge (GCD), electrochemical impedance spectroscopy (EIS), and long-term cycling on an electrochemical workstation (IM6, Zahner, Kronach, Germany) at room temperature. The configuration, potential window, and electrolyte were set as a three-electrode system, 0–0.5 V and 2 M KOH, respectively. The $Ni_{1-x}Co_x(OH)_2$/NF served as a working electrode directly while Pt net and Ag/AgCl (3.5 M KCl) were employed as a counter electrode and reference electrode. The measured ranges of scan rate for CV, current density for GCD, and frequency for EIS were 2–50 mV/s, 1–20 A/g and 100 mHz–100 kHz, respectively. The long-term cycling was executed at a constant current density of 20 A/g for 2000 cycles.

The specific capacitance, energy density, and power density based on GCD measurement are defined as Equations (1)–(3), respectively [43]:

$$C = \frac{It}{mV}, \tag{1}$$

$$E = \frac{CV^2}{7.2}, \tag{2}$$

$$P = \frac{3.6E}{t}. \tag{3}$$

where, $C$ (F/g) is the specific capacitance, $I$ (A) is the discharge current, $\Delta t$ (s) is the discharge time, $m$ (g) is the mass of active material, $\Delta V$ (V) is the voltage window, $E$ (Wh/kg) is the energy density, and $P$ (kW/kg) is the power density.

## 3. Results

By controlling the initial ratio of Ni and Co in the reactants, the binary material experienced an evident morphological evolution from nanosheet to nanowire, investigated in detail by the SEM, as shown in Figure 1b–f. Figure 1a presents the bare nickel foam with a continuous porous three-dimension (3D) network. At a Ni:Co ratio of 9:1, the nanosheets were observed, as shown in Figure 1b. Partial nanosheets intersected among each other, displaying an ambiguous nanoflower-like feature. By further increasing content of Co, the morphology of the composite evolved into nanosheet completely, as shown in Figure 1c. Particularly, as more cobalt ions were devoted (intermediate Ni:Co ratio), nanowires appeared and were encapsulated around the nanosheets, as shown in Figure 1d. The constructed integrated geometry provided a large accessible surface area compared with sole nanosheet morphology. At a greater Co to Ni ratio, the nanosheet structures were replaced by uniform nanowires, as shown in Figure 1e. Finally, when Co became dominant (Ni:Co ratio of 1:9), it could be seen that sectional nanowires trended to aggregate, as depicted in Figure 1f. The distinct morphological evolution is ascribed to the competition of Ni and Co cations for hydroxide radicals [44].

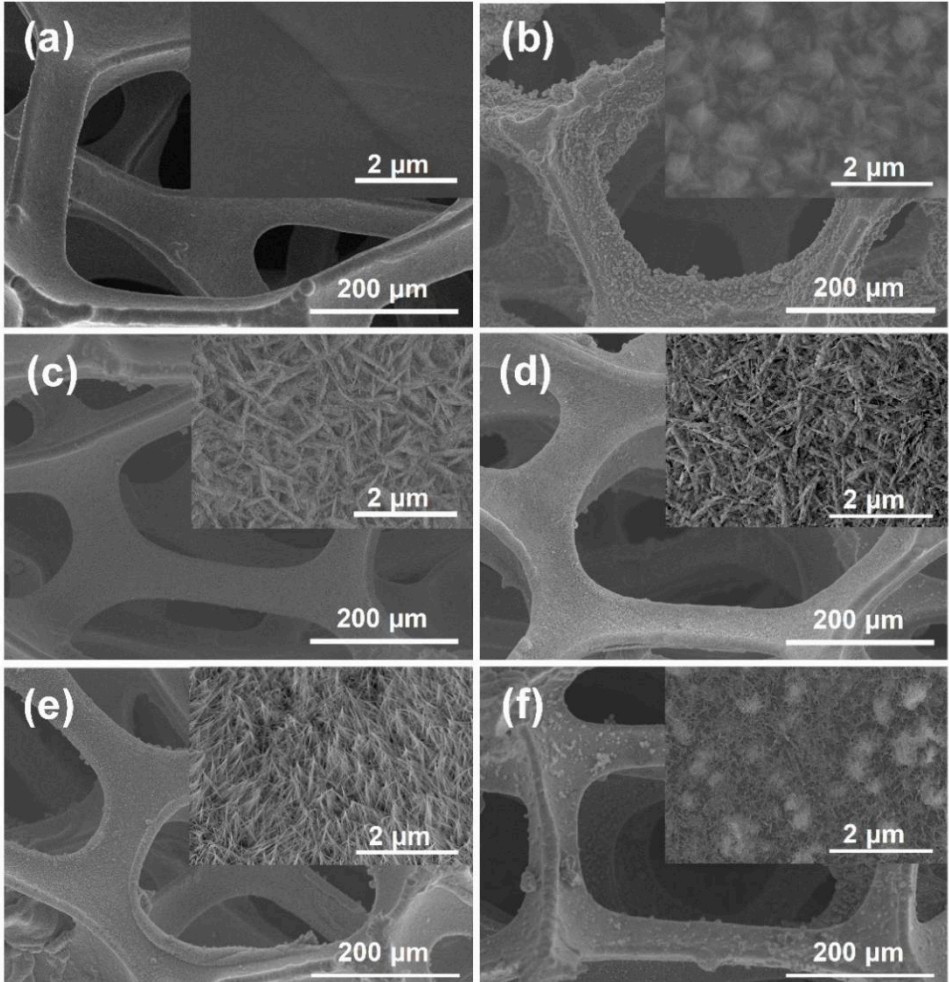

**Figure 1.** Scanning electron microscope (SEM) images of the samples at different Ni:Co ratios: (**a**) bare nickel foam; (**b**) 9:1; (**c**) 7:3; (**d**) 5:5; (**e**) 3:7; (**f**) 1:9.

The XRD patterns of as-prepared samples under different Ni:Co ratios are shown in Figure 2. The indexed planes of the diffraction peaks at corresponding 2θ values can be assigned to $Ni(OH)_2$ (JCPDS No. 14-0117) and $Co(OH)_2$ (JCPDS No. 30-0443). The analogous peak positions are believed to be caused by the similar physical and chemical characteristics of Ni and Co [45]. When a larger Co source was applied, the peak signals became weaker and a lacking of partial peaks appeared, which is ascribed to the low crystallinity of the sample [46]. The XPS was carried out to verify the element valence state of the as-synthesized material. The typical XPS survey spectrum of the sample at a Ni:Co ratio of 5:5 is depicted in Figure 3a, where C, Ni, Co, and O were visible. The C element is due to the air exposure of the sample and can be referenced to calibrate the binding energy [47,48]. In the high resolution XPS spectrum of Ni 2p, shown in Figure 3b, the peaks of Ni $2p_{3/2}$ and Ni $2p_{1/2}$ at binding energies of 855.6 and 873.5 eV (energy separation of 17.9 eV), as well as two obvious shakeup satellites (denoted as Sat.), indicate the characteristics of $Ni^{2+}$ [49]. Figure 3c illustrates the core-level XPS spectrum of Co 2p, the peaks at binding energies of 781.4 and 796.9 eV standing for Co $2p_{3/2}$ and Co $2p_{1/2}$ (energy gap of 15.5 eV), identify Co as $Co^{2+}$ [50,51]. The single peak located at 531.1 eV in the O 1s spectrum, shown in Figure 3d, is assigned to OH- [48,52]. Hence, it can be concluded that Ni-Co double hydroxides were successfully formed under the present experimental conditions.

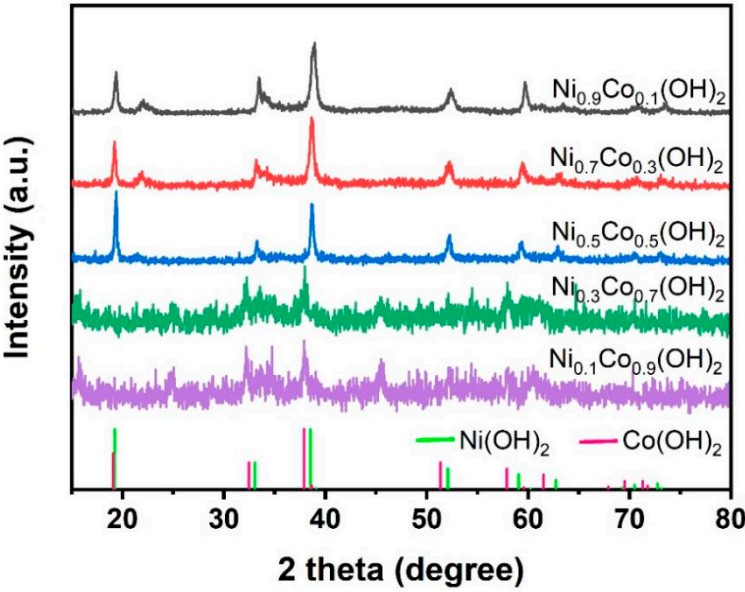

**Figure 2.** X-ray powder diffraction (XRD) patterns of the $Ni_{1-x}Co_x(OH)_2$.

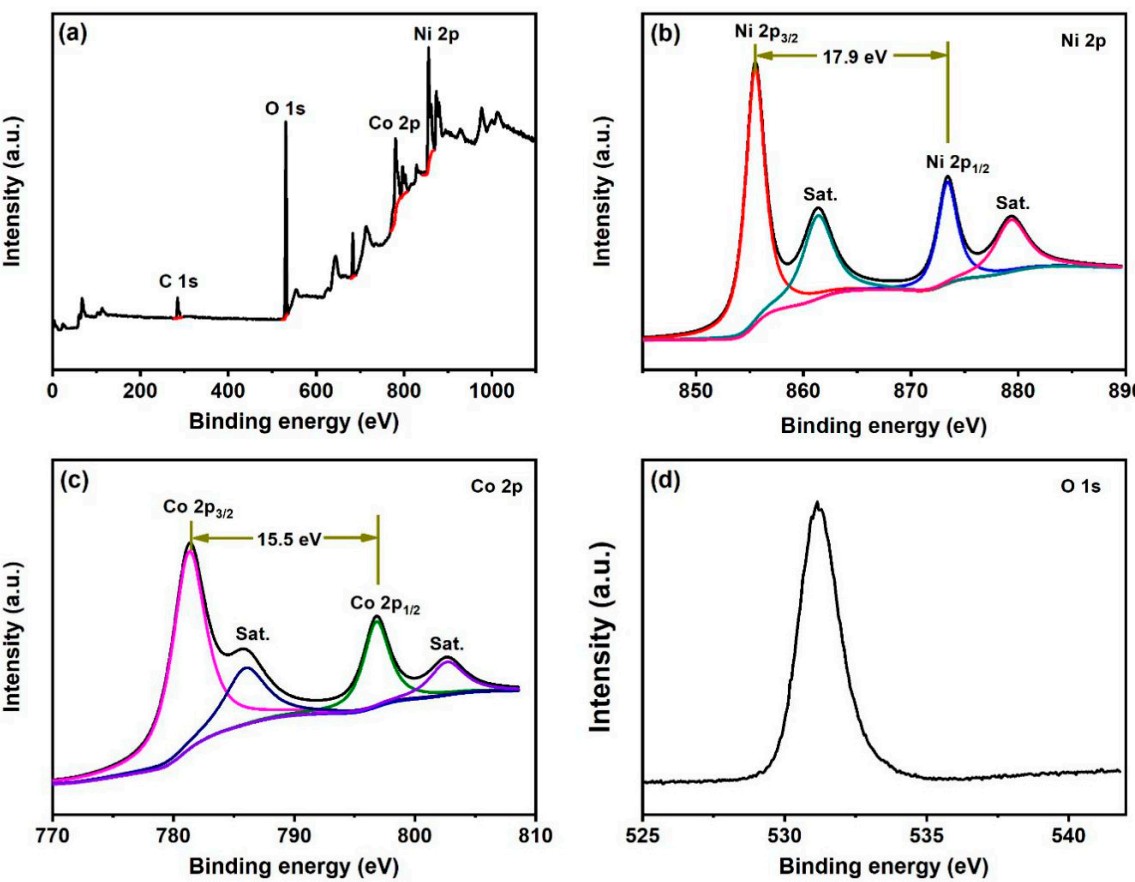

**Figure 3.** X-ray photoelectron spectroscopy (XPS) spectra of the $Ni_{0.5}Co_{0.5}(OH)_2$: (**a**) survey spectrum; (**b**) Ni 2p; (**c**) Co 2p; (**d**) O 1s.

The CV curves were initially characterized in a potential window of 0–0.5 V (vs. Ag/AgCl) to evaluate the supercapacitor behavior of the binary materials. The $Ni_{1-x}Co_x(OH)_2$/NF was directly employed as the binder-free working electrode in a typical three-electrode configuration. The CV curves of the active materials under different Ni:Co ratios, measured at a scan rate of 2 mV/s, are shown

in Figure 4a. All CV curves present a distinct pair of peaks, which involves three reversible faradaic redox processes, expressed as Equations (4)–(6) [41]:

$$Ni(OH)_2 + OH^- \leftrightarrow NiOOH + H_2O + e^-, \tag{4}$$

$$Co(OH)_2 + OH^- \leftrightarrow CoOOH + H_2O + e^-, \tag{5}$$

$$CoOOH + OH^- \leftrightarrow CoO_2 + H_2O + e^-. \tag{6}$$

The merged and indistinguishable redox peaks reveal the mixed uniformity of Ni and Co in the binary hydroxides [53]. Further, the peaks in anodic sweep move to lower potential when Co content increases, because the potential of Co transition is lower than that of Ni. In other words, the oxidation peak corresponding to Ni is closer to the voltage window limit [44,54]. The potential window is mainly determined according to the range where effective faradaic reactions occur. At a defined scan rate, the specific capacitance is correlated with the curvilinear integrated area positively based on CV evaluation [43]. Clearly, the maximum area of the CV curve is achieved at a Ni:Co ratio of 5:5. Figure 4b displays the CV curves of the $Ni_{0.5}Co_{0.5}(OH)_2$ electrode, tested at scan rates of 2, 5, 10, 20, and 50 mV/s. The coupled redox peaks in the CV curves shift positively or negatively with increased scan rates. The classical phenomenon also confirms the pseudocapacitive property of the obtained material [55]. Besides, the relationship of peak currents and the square root of the scan rates delivers a linear response with a steep slope, as described in the inset of Figure 4b. The distortion of the CV curves, obtained from 2 to 50 mV/s, is almost negligible. The characteristics above illustrate the excellent reversibility and rapid response of the $Ni_{0.5}Co_{0.5}(OH)_2$ electrode, promisingly leading to a desirable rate capability [56].

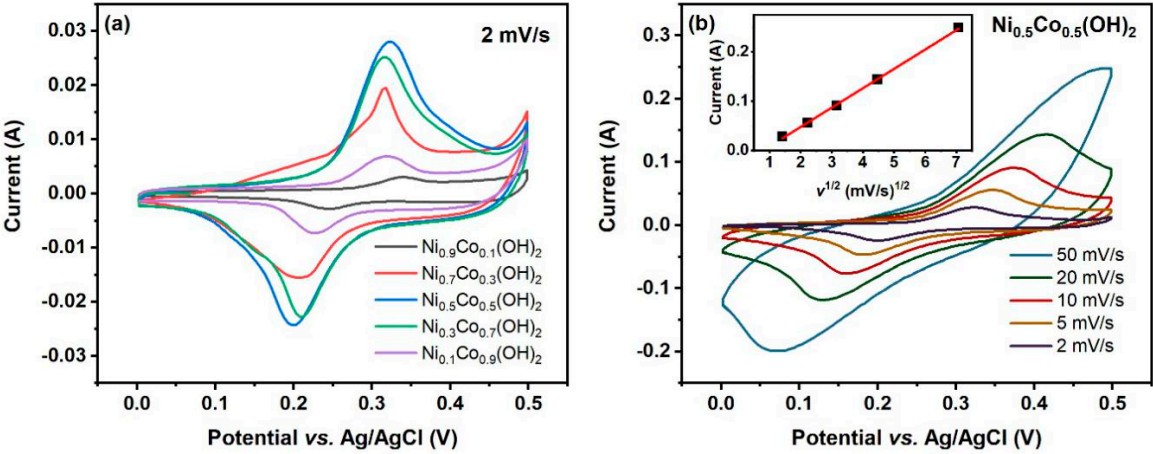

**Figure 4.** (**a**) Cyclic voltammetry (CV) curves of the $Ni_{1-x}Co_x(OH)_2$ electrodes at scan rate of 2 mV/s. (**b**) CV curves of the $Ni_{0.5}Co_{0.5}(OH)_2$ electrode at scan rates from 2 to 50 mV/s.

The GCD tests through the same three-electrode system were conducted on the binary hydroxide electrodes to demonstrate the electrochemical properties further. The GCD curves of the $Ni_{1-x}Co_x(OH)_2$ electrodes, measured at current densities of 1, 2, 5, 10, and 20 A/g in a potential range of 0–0.5 V (vs. Ag/AgCl), are shown in Figure 5a–e, respectively. As expected, all non-linear curves deliver well-defined plateaus, arising from redox reactions, also suggesting the pseudocapacitive features of the synthesized materials [54]. What is more, the plateaus are still dimly visible, even at a current density of 20 A/g, which implies a slight decay of the specific capacitance at high current density could possibly be reached. In GCD measurements, the specific capacitance is proportional to the discharge time [43]. Figure 5f intuitively depicts that the proposed $Ni_{0.5}Co_{0.5}(OH)_2$ electrode takes the longest time to complete one discharge process, which is in good agreement with the CV results exhibited in Figure 4a.

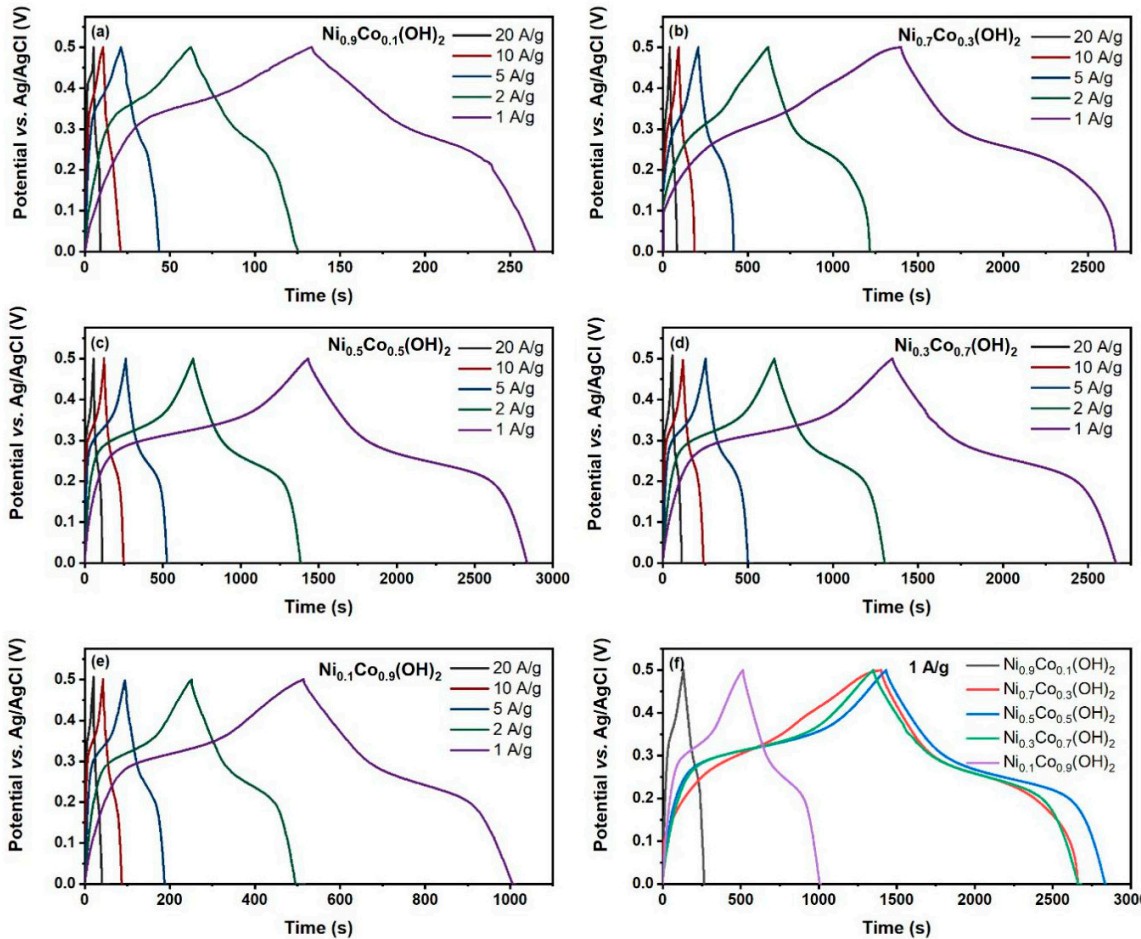

**Figure 5.** Galvanostatic charge/discharge (GCD) curves at current densities from 1 to 20 A/g:
(**a**) $Ni_{0.9}Co_{0.1}(OH)_2$; (**b**) $Ni_{0.7}Co_{0.3}(OH)_2$; (**c**) $Ni_{0.5}Co_{0.5}(OH)_2$; (**d**) $Ni_{0.3}Co_{0.7}(OH)_2$;
(**e**) $Ni_{0.1}Co_{0.9}(OH)_2$; (**f**) GCD curves of the $Ni_{1-x}Co_x(OH)_2$ electrodes at current density of 1 A/g.

The specific capacitances based on the GCD tests were determined according to Equation (1).
The calculated values as a function of current densities for all as-prepared electrodes are plotted in
Figure 6a. All of the function graphs show a similar attenuation tendency, namely, the produced specific
capacitances decrease accompanied by the boost of current densities. Admittedly, the unsatisfying fade
is inevitable, owing to the limited diffusion of electrolyte ions (OH- in this work). Briefly, in comparison
with low current density, the time for OH- transfer is inadequate at high current density, leading to the
fact that involved active material in redox reactions is also insufficient [57]. The specific capacitance
becomes greater with the content of Co in the binary hydroxide until the intermediate ratio. Beyond this
ratio, a decrease in capacitive performance appears. This phenomenon is consistent with previous
literature [54,58–61]. The $Ni_{0.5}Co_{0.5}(OH)_2$ electrode delivers high specific capacitances of 2807, 2751,
2622, 2444, and 2222 F/g at current densities of 1, 2, 5, 10 and 20 A/g, which is attributed to the integrated
nanosheet/nanowire geometry and broadened redox behavior at a Ni:Co ratio of 5:5. The specific
capacitance, reached as an important parameter for supercapacitors, manifests in the great application
prospect of the obtained $Ni_{0.5}Co_{0.5}(OH)_2$ electrode. Among five composites, the $Ni_{0.9}Co_{0.1}(OH)_2$ and
$Ni_{0.1}Co_{0.9}(OH)_2$ electrodes express large differences in specific capacitances compared with the other
three samples. The poor ability under the same measured conditions is mainly due to the aggregation of
material, which deteriorates the accessibility of electrolyte ions in electroactive material. Furthermore,
the $Ni_{0.5}Co_{0.5}(OH)_2$ electrode still maintains a specific capacitance of 79.2%, even when the current
density increases 20-times more than the initial value, representing an excellent rate capability. What is
noteworthy is that the $Ni_{0.3}Co_{0.7}(OH)_2$ electrode offers a slightly higher rate capability of 82% (2624 and

2151 F/g at 1 and 20 A/g) than that of the $Ni_{0.5}Co_{0.5}(OH)_2$ sample, which highlights the advantage of nanowire morphology as a one-dimension (1D) geometry, which can better guarantee effective ion diffusion and electron transportation at high current density [44,62]. Based on Equations (2) and (3), the Ragone plots of the samples were obtained to further illustrate the electrochemical properties, as exhibited in Figure 6b. The $Ni_{0.5}Co_{0.5}(OH)_2$ sample shows energy densities of 97.5, 95.5, 91, 84.9 and 77.2 Wh/kg at power densities of 0.25, 0.5, 1.25, 2.5 and 5 kW/kg, respectively.

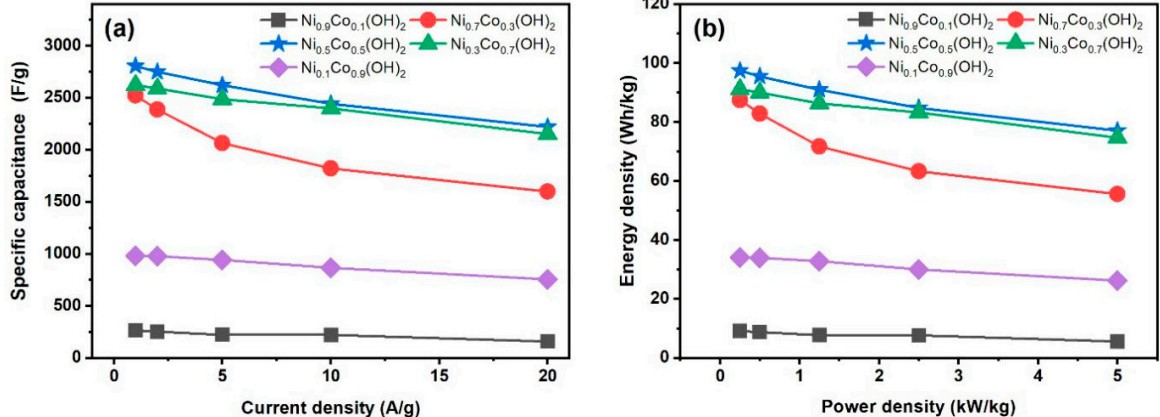

**Figure 6.** (**a**) Specific capacitances as a function of current densities and (**b**) Ragone plots of the $Ni_{1-x}Co_x(OH)_2$ samples.

The EIS studies over the frequency range of 100 mHz to 100 kHz in 2 M KOH were carried out on $Ni_{0.7}Co_{0.3}(OH)_2$, $Ni_{0.5}Co_{0.5}(OH)_2$, and $Ni_{0.3}Co_{0.7}(OH)_2$ as optimized electrodes. Figure 7a records the typical Nyquist plots. Generally, the intrinsic resistances (electrolyte and material) and contact resistances (electrolyte/material and material/current collector) are summarized as $R_s$, which is valued from the intersection of the EIS plot and real axis. Figure 7b illustrates the high frequency in an enlarged view, in which the semicircle corresponds to the charge transfer resistance ($R_{ct}$). The straight line at the low frequency portion represents the ion diffusion in the electrolyte, arising in a Warburg element ($W$) [61,63,64]. The $Ni_{0.3}Co_{0.7}(OH)_2$ electrode delivers the smallest diameter (expressing lowest $R_{ct}$) and largest slop (suggesting fastest ion diffusion), which is consistent with its outstanding rate performance. The highest specific capacitance produced by the $Ni_{0.5}Co_{0.5}(OH)_2$ electrode is due to: (1) the union of Ni and Co at ratio of 5:5 generating most electroactive sites from the feasible interaction of the valence state electron [54]; (2) the $R_s$ value of the $Ni_{0.5}Co_{0.5}(OH)_2$ electrode being lower than the others. Overall, the EIS spectra well coincide with the aforementioned CV and GCD results. Figure 7c presents the fitting equivalent circuit, where $C_{dl}$ and $C_{ps}$ account for double-layer capacitance and pseudocapacitance, respectively.

The long-term electrochemical stability was assessed via repetitive GCD tests at a constant current density of 20 A/g. Figure 8 displays the capacitance retentions and coulombic efficiencies of the $Ni_{0.5}Co_{0.5}(OH)_2$ electrode as a function of the cycle numbers (the inset depicts the GCD curves of the last 10 cycles). The coulombic efficiency is calculated based on Equation (7) [63]:

$$\eta = \frac{t_d}{t_c},\tag{7}$$

where, $\eta$ (%) is the coulombic efficiency, $t_d$ (s) is the discharge time, and $t_c$ (s) is the charge time. As recorded in Figure 8, the deduced capacitance retentions gradually decreased at first, and after approximately 1000 cycles remained nearly constant. Impressively, even after 2000 cycles, the $Ni_{0.5}Co_{0.5}(OH)_2$ electrode still reached 80.2% of its initial specific capacitance and its high coulombic efficiency exceeded 90%. The capacitance loss after long-term cycling is inevitable, which is likely attributed to several reasons, (e.g., the presence of irreversible redox reactions, damage to the electrode,

or impurities in the electrolyte) [65]. The remarkable durability performance of the $Ni_{0.5}Co_{0.5}(OH)_2$ electrode is pivotal for potential commercial application. In contrast, the cycling stability of the $Ni_{0.3}Co_{0.7}(OH)_2$ electrode is poor (capacitance retention of 62.1% after 2000 cycles), possibly since partial nanowires were not attached to the nickel foam tightly enough, giving rise to low material utilization for capacitance.

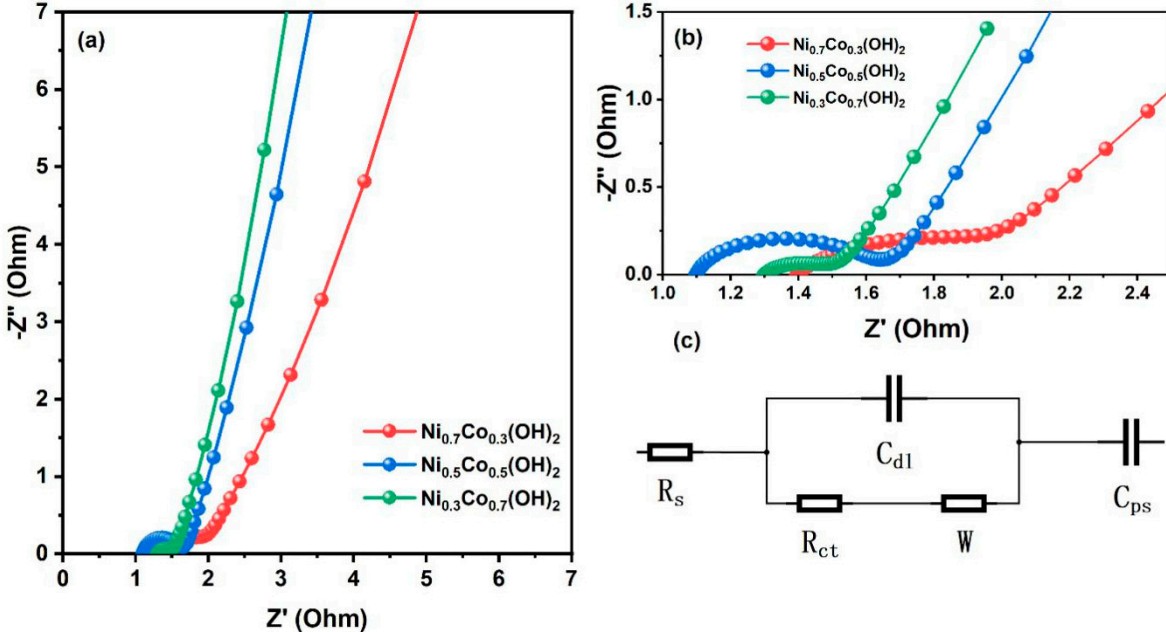

**Figure 7.** (**a**) Electrochemical impedance spectroscopy (EIS) plots of the $Ni_{0.7}Co_{0.3}(OH)_2$, $Ni_{0.5}Co_{0.5}(OH)_2$, and $Ni_{0.3}Co_{0.7}(OH)_2$ electrodes. (**b**) Enlarged view of the high frequency region. (**c**) Fitting equivalent circuit.

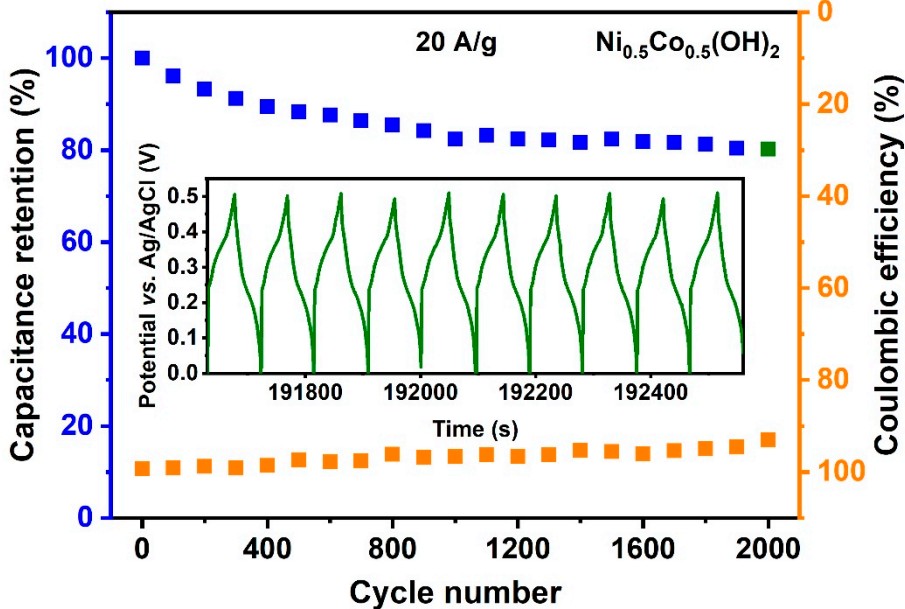

**Figure 8.** Cycling performance and coulombic efficiency of the $Ni_{0.5}Co_{0.5}(OH)_2$ electrode.

## 4. Discussion

To summarize, the $Ni_{1-x}Co_x(OH)_2$ electrode possessed the best overall supercapacitor performance when x equals 0.5 was investigated in terms of specific capacitance, rate capability, and cycling stability. These superior behaviors are attributed to the common advantages of the binary system and the unique

merits of $Ni_{0.5}Co_{0.5}(OH)_2$. Firstly, the synthesized material is grown directly onto the surface of nickel foam rather than by virtue of a binder, affording an intimate electric contact. Secondly, the oxidation peak of Ni moves towards comparatively lower potential by introducing Co, which avoids possible solvent oxidation [44]. Thirdly, the overlapping redox peaks broaden the redox features and improve redox response. In addition, the integrated nanosheet/nanowire architecture of $Ni_{0.5}Co_{0.5}(OH)_2$ increases the active sites for redox reaction and pathways for electron transportation. On the other hand, the $Ni_{0.5}Co_{0.5}(OH)_2$ electrode delivers the lowest intrinsic and contact resistances (i.e., $R_s$). Moreover, the $Ni_{0.5}Co_{0.5}(OH)_2$ possibly possesses the most electroactive sites generated by valence interchange or charge hopping between cations [58]. The specific capacitance of the $Ni_{0.5}Co_{0.5}(OH)_2$ electrode is even higher than those of some hybrid materials [66–68]. Apart from remarkable specific capacitance, the high rate capability of 79.2% at current density of 20 A/g and the slight capacitance decay of 19.8% after 2000 cycles are also satisfied.

## 5. Conclusions

In summary, the Ni-Co binary hydroxide system was fabricated by a facile one-step hydrothermal reaction. By tailoring the cation ratio in preparation, the morphology of the binary system evolved from nanosheet to nanowire straightforwardly, as a consequence, giving rise to different supercapacitor behaviors. The optimal electrode achieved at a Ni:Co ratio of 5:5 exhibited a prominent specific capacitance of 2807 F/g at a current density of 1 A/g (based on active material) as well as outstanding rate (79.2% capacitance retention at 20 A/g) and cycling (80.2% capacitance retention after 2000 cycles) performances. The remarkable criteria and the environmentally friendly method manifest great prospects of the obtained $Ni_{0.5}Co_{0.5}(OH)_2$ electrode for commercial application in high-performance supercapacitors. Furthermore, this novel approach is also promisingly adopted to develop other binary systems using different electrochemically active metal hydroxides for wide applications not only in supercapacitors, but also in catalysts, sensors, and so forth.

**Author Contributions:** Conceptualization, X.C. and P.O.; methodology, X.F., P.O., and X.C.; software, X.F.; validation, X.F., P.O., and X.C.; formal analysis, X.F.; investigation, X.C. and P.O.; resources, X.C. and P.O.; data curation, X.F., P.O., and X.C.; writing—original draft preparation, X.F.; writing—review and editing, X.C. and P.O.; visualization, X.F., P.O., and X.C.; supervision, X.C. and P.O.; project administration, X.C. and P.O.; funding acquisition, X.C. All authors have read and agreed to the published version of the manuscript.

**Funding:** This research was funded by the Research Council of Norway (RCN, grant number 221860/F60) and the Norwegian Micro- and Nano-Fabrication Facility (NorFab, project number 245963). Xiao Fan was financially supported by the China Scholarship Council (CSC, grant number 201506930018).

**Acknowledgments:** The authors gratefully acknowledge lab engineers Zekija Ramic, Thomas Martinsen, Tayyib Muhammad, Anh Tuan Thai Nguyen, and Birgitte Kasin Hønsvall for their kind help associated with this work. The authors also specially thank Einar Halvorsen, Pai Lu, and Yongjiao Sun for their critical suggestions.

**Conflicts of Interest:** The authors declare no conflict of interest.

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
