# Peer review of "One-Step and Morphology-Controlled Synthesis of Ni-Co Binary Hydroxide on Nickel Foam for High-Performance Supercapacitors"

_applsci, doi:10.3390/app10113814_

Round 1

Reviewer 1 Report

The authors described Ni-Co binary system for pseudosupercapacitor.

Readers may be interested in Ni or Co itself as electrodes.

They may include data for Ni 100% and Co 100 % system for comparison.

Any detail reason for mixing system could be added more.

Thanks.

Author Response

The authors described Ni-Co binary system for pseudosupercapacitor. Readers may be interested in Ni or Co itself as electrodes. They may include data for Ni 100% and Co 100 % system for comparison. Any detail reason for mixing system could be added more. Thanks.

Response: Owing to stronger layered orientation, reduced resistance, increased active sites generated by valence interchange or charge hopping between cations, synergistic redox reaction, moderate redox potential and decreased redox peak potential difference [1-5], the Ni-Co binary system outperforms the corresponding single hydroxide, which is extensively acknowledged and verified. The detail reasons were added in Introduction of the revised manuscript and five representative literatures were cited. In this study, our work mainly focus on addressing the issues like poor overall supercapacitors performance and complex synthesis method in previous achievements of Ni-Co binary system.     

[1]   Liu, X.; Ma, R.; Bando, Y.; Sasaki, T. A general strategy to layered transition‐metal hydroxide nanocones: tuning the composition for high electrochemical performance. Adv. Mater. 201224, 2148-2153.

[2]   Armstrong, R.D.; Charles, E.A. Some effects of cobalt hydroxide upon the electrochemical behaviour of nickel hydroxide electrodes. J. Power Sources 198925, 89-97.

[3]    Nguyen, T.; Boudard, M.; Carmezim, M.J.; Montemor, M.F. Layered Ni (OH)2-Co (OH)2 films prepared by electrodeposition as charge storage electrodes for hybrid supercapacitors. Sci. Rep. 20177, 1-10.

[4]  Chen, J.C.; Hsu, C.T.; Hu, C.C. Superior capacitive performances of binary nickel–cobalt hydroxide nanonetwork prepared by cathodic deposition. J. Power Sources 2014253, 205-213.

[5]   Wang, C.; Zhang, X.; Xu, Z.; Sun, X.; Ma, Y. Ethylene glycol intercalated cobalt/nickel layered double hydroxide nanosheet assemblies with ultrahigh specific capacitance: structural design and green synthesis for advanced electrochemical storage. ACS Appl. Mater. Interfaces 20157, 19601-19610.

Reviewer 2 Report

This is an experimental study of fabricating Ni-Co binary hydroxide grown on nickel foam through a one-step process for pseudocapacitive electrode application. The morphology of the fabricated binary hydroxide was controllable by tuning the Ni/Co ratio. In systematical electrochemical measurements, the prepared binary material on nickel foam could be employed as binder-free working electrode directly. The capacitance loss is 19.8% after 2000 cycles, demonstrating good long-term cyclic stability. The outstanding supercapacitors behaviors are benefited from unique structure and synergistic contributions, indicating potential of the obtained binary hydroxide electrode for high-performance energy storage devices. I recommend publication subject to the following two improvements:

  1. I am not sure about the novelty of this work. hydroxides are already used as batteries for decades. The authors should explain better what is new here.
  2. I recommend citing the large amount of literature on these materials including https://doi.org/10.1002/cctc.201902289, https://doi.org/10.1016/j.electacta.2018.03.075. 

Author Response

This is an experimental study of fabricating Ni-Co binary hydroxide grown on nickel foam through a one-step process for pseudocapacitive electrode application. The morphology of the fabricated binary hydroxide was controllable by tuning the Ni/Co ratio. In systematical electrochemical measurements, the prepared binary material on nickel foam could be employed as binder-free working electrode directly. The capacitance loss is 19.8% after 2000 cycles, demonstrating good long-term cyclic stability. The outstanding supercapacitors behaviors are benefited from unique structure and synergistic contributions, indicating potential of the obtained binary hydroxide electrode for high-performance energy storage devices. I recommend publication subject to the following two improvements.

  1. I am not sure about the novelty of this work. Hydroxides are already used as batteries for decades. The authors should explain better what is new here.

Response: In pseudocapacitors research field, RuO2 is the most explored electrode material. Unfortunately, the high cost and environment harmfulness severely limit its applications. Therefore, it is necessary to seek cost-effective and environmentally friendly materials as alternatives. Ni(OH)2 and Co(OH)2 sparked considerable interest in last decade owing to their low cost, abundant resources and more importantly their superior theoretical specific capacitances. In recent years, binary Ni-Co hydroxide, which have been shown to outperform the corresponding single hydroxide due to a series of advantages such as increased active sites, improved redox response, reduced resistance and so forth, was triggered a great deal of efforts. However, the progresses up to now fail to exhibit satisfied overall supercapacitors performances mainly because of undesirable morphology and involved binder. This work thus focuses on overcoming the drawbacks. The fabricated Ni-Co binary hydroxide electrode displays excellent overall supercapacitors performances with respect to loading mass, specific capacitance, rate capability and cycling stability. The one-step synthesized method further manifests a promising application in commercial field.

  1. I recommend citing the large amount of literature on these materials including https://doi.org/10.1002/cctc.201902289, https://doi.org/10.1016/j.electacta.2018.03.075.

Response: The suggested literatures were cited in the revised manuscript.

Reviewer 3 Report

The manuscript described a method to anchored Ni-Co nanoparticles on Ni foam using hydrothermal process. Full materials characterization is presented, and the electrochemical performance of the electrode in supercapacitors is evaluated. I would recommend publishing after considering the following comment.
The authors described a method to control the composition by controlling the Ni/Co ions in the initial charge. However, this concept seems to ignore the oxidation of the Ni foam. Can the authors’ comments or explain why they have not included nickel hydroxide from the foam.
The authors need to elaborate more on the reason of why 1:1 ratio gave the highest capacitance
2000 cycles are not high in supercapacitors. The authors need to elaborate more on the reason why the cell lost almost 20% of the capacitance in 2000 cycles

Author Response

The manuscript described a method to anchored Ni-Co nanoparticles on Ni foam using hydrothermal process. Full materials characterization is presented, and the electrochemical performance of the electrode in supercapacitors is evaluated. I would recommend publishing after considering the following comment.

The authors described a method to control the composition by controlling the Ni/Co ions in the initial charge. However, this concept seems to ignore the oxidation of the Ni foam. Can the authors’ comments or explain why they have not included nickel hydroxide from the foam. The authors need to elaborate more on the reason of why 1:1 ratio gave the highest capacitance. 2000 cycles are not high in supercapacitors. The authors need to elaborate more on the reason why the cell lost almost 20% of the capacitance in 2000 cycles.

Response: To the best of our knowledge, the direct growth of nickel hydroxide from nickel foam without adding any additional nickel sources is usually controlled in strong alkaline solution [1,2]. Considering the involved nickel source in synthesis process and achieved relatively high loading mass of 1.5 mg/cm2, we suggest a negligible contribution of nickel hydroxide from the foam. In this study, the Ni0.5Co0.5(OH)2 sample shows the highest specific capacitance. Similar phenomena were reported previously [3,4]. It is mainly ascribed to three advantages: (1) compared with other samples, the Ni0.5Co0.5(OH)2 delivers lowest intrinsic and contact resistances, i.e. Rs; (2) the unique integrated nanosheet/nanowire morphology could provide increased active sites and numerous pathways;  (3) the binary system at Ni/Co ratio of 1:1 possibly generates most electroactive sites owing to valence interchange or charge hopping between cations [5] (added in the revised manuscript at line 259-263). The capacitance loss after long-term cycling is inevitable, which may be attributed to several possible reasons, e.g. the presence of irreversible redox reactions, the damage of electrode, or the impurities in electrolyte [6] (added in the revised manuscript at line 238-240).

[1]    Xiong, X.H.; Wang, Z.X.; Guo, H.J.; Li, X.H. Facile synthesis of ultrathin nickel hydroxides nanoflakes on nickel foam for high-performance supercapacitors. Mater. Lett. 2015138, 5-8.

[2]    Xiong, X.; Ding, D.; Chen, D.; Waller, G.; Bu, Y.; Wang, Z.; Liu, M. Three-dimensional ultrathin Ni(OH)2 nanosheets grown on nickel foam for high-performance supercapacitors. Nano Energy 201511, 154-161.

[3]  Xie, L.; Hu, Z.; Lv, C.; Sun, G.; Wang, J.; Li, Y.; He, H.; Wang, J.; Li, K. CoxNi1−x double hydroxide nanoparticles with ultrahigh specific capacitances as supercapacitor electrode materials. Electrochim. Acta 201278, 205-211.

[4]  Hu, Z.A.; Xie, Y.L.; Wang, Y.X.; Wu, H.Y.; Yang, Y.Y.; Zhang, Z.Y. Synthesis and electrochemical characterization of mesoporous CoxNi1−x layered double hydroxides as electrode materials for supercapacitors. Electrochim. Acta 200954, 2737-2741.

[5]   Cheng, Y.; Zhang, H.; Varanasi, C.V.; Liu, J. Improving the performance of cobalt–nickel hydroxide-based self-supporting electrodes for supercapacitors using accumulative approaches. Energy Environ. Sci. 20136, 3314-3321.

[6]   Lu, P.; Ohlckers, P.; Müller, L.; Leopold, S.; Hoffmann, M.; Grigoras, K.; Ahopelto, J.; Prunnila, M.; Chen, X. Nano fabricated silicon nanorod array with titanium nitride coating for on-chip supercapacitors. Electrochem. Commun. 201670, 51-55.

Reviewer 4 Report

The paper is fundamentally interesting and should be published.  There are some major technical issues that must be resolved before publication. 

  1. The authors must provide an actual picture/cartoon/whatever of the physical test arrangement.  This reviewer is puzzled about basic issues:  How is a connection made between an oxide coated electrode and a galvanostat?  Is the electrode submerged in a liquid electrolyte?  The authors must precisely describe the chemistry of this electrolyte.  What changes occur if a different electrolyte is employed?
  2.  Why do the authors insist on employing F/g?  It is quite clear from the missing 'flat' in the cyclic voltametry data and the wildly non-linear discharge and charge curves from the galvanostat that the 'capacitance' is voltage dependent.  Hence, providing a 'capacitance' is some sort of wildly misleading 'global average'.  Unacceptable.
  3. Could the authors present the data as energy density?  This is a better means to make an 'apples to apples' comparison with other capacitors, particularly as super capacitors are not used as circuit elements, but rather for energy storage or power. No one is interested in a high 'capacitance' device with a top voltage of 0.4 V because the energy density is low.  Better: Plot energy density vs. discharge time. A clear pattern will emerge.
  4. Why is the maximum voltage 0.5 V?
  5. The high frequency data is obscured,  and even if clarification is provided, it is irrelevant as super capacitors never work at high frequency.
  6. The standard commercial process for rating capacitors generally involves holding the capacitor at maximum voltage for up to an hour before discharge.  Why isn't that done here?  And why do long hold times lead to performance enhancement according to 'pseudo capacitance' theory?
  7. Have the authors considered that the entire 'pseudo capacitance' theory is fundamentally wrong?  Have they considered applying super dielectric materials theory to their results?

Author Response

The paper is fundamentally interesting and should be published. There are some major technical issues that must be resolved before publication.

  1. The authors must provide an actual picture/cartoon/whatever of the physical test arrangement. This reviewer is puzzled about basic issues: How is a connection made between an oxide coated electrode and a galvanostat? Is the electrode submerged in a liquid electrolyte? The authors must precisely describe the chemistry of this electrolyte. What changes occur if a different electrolyte is employed?

Response: In this study, three-electrode configuration was employed in electrochemical measurements, in which the Ni-Co binary hydroxide grown on nickel foam was used as working electrode and the Pt net and Ag/AgCl were served as counter electrode and reference electrode. All electrodes were submerged in KOH solution. In GCD test, a constant charge current was applied on the working electrode until the potential reached to set value, then a constant discharge current (equal magnitude of the charge current) was conducted the working electrode until the potential backed to original value [1]. The schematic of three-electrode configuration was shown in Figure 1.

Figure 1. Schematic of three-electrode configuration.

The electrolyte ensures the reversible redox reactions, as expressed by Equation (1), (2) and (3) [2]. Hence, alkaline electrolyte should be employed, in which KOH is representative and commonly used thanks to its high conductivity and low cost [3-5]. The schematic diagram and the Equations of redox reactions were added to the revised manuscript.

(1)

(2)

(3)

[1]  Benjamin H. Materials Synthesis and Characterization for Micro-supercapacitor Applications. Ph.D. Dissertation 2013, UC Berkeley, USA.

[2]    Zhang, J.; Cheng, J.P.; Li, M.; Liu, L.; Liu, F.; Zhang, X.B. Flower-like nickel–cobalt binary hydroxides with high specific capacitance: tuning the composition and asymmetric capacitor application J. Electroanal. Chem. 2015743, 38-45.

[3]  González, A.; Goikolea, E.; Barrena, J.A.; Mysyk, R. Review on supercapacitors: technologies and    materials. Renew. Sust. Energ. Rev. 201658, 1189-1206.

[4]   Wang, Y.; Song, Y.; Xia, Y. Electrochemical capacitors: mechanism, materials, systems, characterization and applications. Chem. Soc. Rev. 201645, 5925-5950.

[5]    Pal, B.; Yang, S.; Ramesh, S.; Thangadurai, V.; Jose, R. Electrolyte selection for supercapacitive devices: a critical review. Nanoscale Adv. 20191, 3807-3835.

  1. Why do the authors insist on employing F/g? It is quite clear from the missing 'flat' in the cyclic voltametry data and the wildly non-linear discharge and charge curves from the galvanostat that the 'capacitance' is voltage dependent. Hence, providing a 'capacitance' is some sort of wildly misleading 'global average'. Unacceptable.

Response: In supercapacitors research field, the specific capacitance (F/g is frequently adopted) is the most important parameter in assessment of the electrode. In this study, the peaks in CV curves and the plateaus in GCD curves are ascribed to the redox reactions. In addition, the energy densities vs. power densities was plotted (seen in response of comment 3).

  1. Could the authors present the data as energy density? This is a better means to make an 'apples to apples' comparison with other capacitors, particularly as super capacitors are not used as circuit elements, but rather for energy storage or power. No one is interested in a high 'capacitance' device with a top voltage of 0.4 V because the energy density is low. Better: Plot energy density vs. discharge time. A clear pattern will emerge.

Response: As the power density is also a key parameter, we suggest a plot of energy densities vs. power densities (usually called Ragone plot) to further illustrate the electrochemical properties, as shown in Figure 2b. The Ni0.5Co0.5(OH)2 sample exhibits energy densities of 97.5, 95.5, 91, 84.9 and 77.2 Wh/kg at power densities of 0.25, 0.5, 1.25, 2.5 and 5 kW/kg, respectively. The Ragone plots were added to the revised manuscript.

Figure 2. (a) Specific capacitances as a function of current densities and (b) Ragone plots of the Ni1-xCox(OH)2 samples.

  1. Why is the maximum voltage 0.5 V?

Response: The potential window was determined based on the electrochemical experiment. When the maximum voltage was beyond 0.5 V, partial active materials peeled off, which could be attributed to the corrosion of current collector or happening of oxygen evolution reaction.

  1. The high frequency data is obscured, and even if clarification is provided, it is irrelevant as super capacitors never work at high frequency.

Response: EIS measurements involves the application of an alternating potential with a wide frequency range and impedance data can be obtained. The imaginary component of impedance against the real component of impedance was plotted (called Nyquist plot) [1,2]. In general, the Nyquist comprises a semicircle in high frequency range and a straight line in low frequency region. Specifically, the intersection value of plot and real axis indicates the intrinsic resistance (electrolyte and material) and contact resistances (electrolyte/material and material/current collector). The diameter of the semicircle reflects the charge transfer resistance and the slop of the line corresponds to the diffusive resistance [3,4]. Further, the fitting equivalent circuit also can be obtained based on EIS measurements.

[1]  Muzaffar, A.; Ahamed, M.B.; Deshmukh, K.; Thirumalai, J. A review on recent advances in hybrid supercapacitors: Design, fabrication and applications. Renew. Sust. Energ. Rev. 2019101, 123-145.

[2]  Benjamin H. Materials Synthesis and Characterization for Micro-supercapacitor Applications. Ph.D. Dissertation 2013, UC Berkeley, USA.

[3]   Wang, Y.; Lei, Y.; Li, J.; Gu, L.; Yuan, H.; Xiao, D. Synthesis of 3D-nanonet hollow structured Co3O4 for high capacity supercapacitor. ACS Appl. Mater. Interfaces 20146, 6739-6747.

[4]  Tang, Y.; Liu, Y.; Yu, S.; Guo, W.; Mu, S.; Wang, H.; Zhao, Y.; Hou, L.; Fan, Y.; Gao, F. Template-free hydrothermal synthesis of nickel cobalt hydroxide nanoflowers with high performance for asymmetric supercapacitor. Electrochim. Acta 2015161, 279-289.

  1. The standard commercial process for rating capacitors generally involves holding the capacitor at maximum voltage for up to an hour before discharge. Why isn't that done here? And why do long hold times lead to performance enhancement according to 'pseudo capacitance' theory?

Response: In this work, three-electrode configuration rather than assembled supercapacitors (stack configuration) was employed to reveal the electrochemical properties. Hence, holding at maximum voltage before discharge that you suggested was not performed. Thanks for your significant suggestion. We would like to take this suggestion in our next step study on full packaged prototype devices.     

  1. Have the authors considered that the entire 'pseudo capacitance' theory is fundamentally wrong? Have they considered applying super dielectric materials theory to their results?

Response: As early as 1971, a new type of electrochemical capacitance was discovered in RuO2, termed as pseudocapacitance. In 1991, Brian Evans Conway described the electrochemical behavior of RuO2 as a transition from electric double layer capacitors to battery. Owing to high specific capacitance, which satisfies the ever-growing demand, pseudocapacitors gradually become the research hotpot. In recent years, numerous materials' pseudocapacitance nature were studied and some novel theories like intrinsic pseudocapacitance and extrinsic pseudocapacitance were proposed. Unfortunately, even though great efforts were devoted, in-situ pseudocapacitive mechanism study has not been reported. Hence, the pseudocapacitive mechanism still remains controversial currently and further research is needed.
